# Integrated Starches and Physicochemical Characterization of Sorghum Cultivars for an Efficient and Sustainable Intercropping Model

**DOI:** 10.3390/plants11121574

**Published:** 2022-06-15

**Authors:** Maw Ni Soe Htet, Honglu Wang, Lixin Tian, Vivek Yadav, Hamz Ali Samoon, Baili Feng

**Affiliations:** 1State Key Laboratory of Crop Stress Biology for Arid Areas, College of Agronomy, Northwest A&F University, Yangling, Xianyang 712100, China; mawni2018071063@nwafu.edu.cn (M.N.S.H.); 2018050099@nwsuaf.edu.cn (H.W.); 2019060003@nwafu.edu.cn (L.T.); 2State Key Laboratory of Crop Cultivation and Farming System in Northwestern Loess Plateau, College of Agronomy, Northwest A&F University, Yangling, Xianyang 712100, China; 3Rice Bio-Park Research Section, Post-Harvest Technology and Food Science Research Division, Department of Agricultural Research, Yezin, Zayarthiri Township, Nay Pyi Taw 15013, Myanmar; 4State Key Laboratory of Crop Stress Biology in Arid Areas, College of Horticulture, Northwest A&F University, Yangling, Xianyang 712100, China; vivekyadav@nwafu.edu.cn; 5PARC-Water and Agricultural Waste Management Institute, Tando Jam 70060, Sindh, Pakistan; hamzsamoon@yahoo.com

**Keywords:** sorghum cultivars, starch, amylose, physicochemical properties, thermal properties, pasting properties, sustainable intercropping model

## Abstract

Sorghum has good adaptation to drought tolerance and can be successfully cultivated on marginal lands with low input cost. Starch is used in many foods and nonfood industrial applications and as a renewable energy resource. Sorghum starches with different amylose contents affect the different physicochemical properties. In this study, we isolated starches from six sorghum varieties (i.e., Jinza 34, Liaoza 19, Jinnuo 3, Jiza 127, Jiniang 2, and Jiaxian) and investigated them in terms of their chemical compositions and physicochemical properties. All the starch granules had regular polygonal round shapes and showed the characteristic “Maltese cross”. These six sorghum starches showed an A-type diffraction pattern. The highest amylose content of starch in Jinza 127 was 26.90%. Jiaxian had a higher water solubility at 30, 70, and 90 °C. From the flow cytometry analysis based on six sorghum starch granules, Liaoza 19 had a larger and more complex granules (particle percentage (P1) = 66.5%). The Jinza 34 starch had higher peak (4994.00 mPa∙s) and breakdown viscosity (4013.50 mPa∙s) and lower trough viscosity (973.50 mPa∙s). Jinnuo 3 had higher onset temperature, peak temperature, conclusion temperature, gelatinization enthalpy, and gelatinization range. The principal component analysis and hierarchical cluster analysis based on classification of different sorghum starches showed that Jiniang 2 and Jinnuo 3 had similar physicochemical properties and most divergent starches, respectively. Our result provides useful information not only on the use of sorghum starches in food and non-food industries but for the great potential of sorghum-based intercropping systems in maintaining agricultural sustainability.

## 1. Introduction

Sorghum (*Sorghum bicolor* (L.) Moench) has been widely grown in China for at least 40–50 centuries in arid, semiarid, and water-logged regions. Although a production zone is wide, the major growing areas are concentrated in the northern and northeastern parts of the country. Chinese sorghum belongs to a unique “kaoliang” group characterized by adaptation to the Chinese environments [1] and used as a raw material for brewing liquor [2]. This genus has high phenolic compounds, which can prevent cancer, diabetes, obesity, dyslipidemia, cardiovascular disease, and digestive tract disease [3], and is an important food stuff, especially for the industrial point of view [4]. Ethanol and bio–fuel are produced from the byproducts of sorghum starch [5]. Nowadays, sorghum research is mainly important because of its great beneficial ingredient for industrial applications and potential crop as a sorghum-based intercropping for agricultural sustainability. Intercropping is a system of growing two or more crops spatially and temporally in the same area, which is very common among small farmers across the world [6,7]. In intercropping systems, both crops almost have the similar growth periods and they required high amount of inputs to produce higher intercrop yields [8,9,10]. Starch, which consists of two macromolecular, namely, amylose and amylopectin, is one of the major chemical components of sorghum grains [1]. The most common amylose and amylopectin ratio is approximately 1:4; however, some plants produce starch with low amylose, known as waxy starch. Starches with more than 40% amylose content are regarded as high amylose starch [11,12].

Corn, wheat, and potato are also different botanical sources of starch, which can be used in many industrial applications due to the fact of starch’s versality [13]. Most researcher have extensively characterized corn, wheat, and potato; however, little interest has been given to the physicochemical properties of sorghum starches. Some studies have reported that the content of amylose can affect the differences in physicochemical, structural, and functional properties [14]. Therefore, starches with different amylose contents have crucial functions in food and non-food industries. Lin et al. [15] and Zhu et al. [16] reported that maize starches with different amylose contents have diverse thermal and functional properties, annealing, and propionylation. The study of proso and foxtail millets with high amylose had higher proportion of amylopectin chain length distribution [17]. Some researchers have also studied the effects of amylose on the physicochemical properties of wheat [18] and rice [19]. In waxy proso millet, the amylose content had highly positively correlated with the retrogradation percentage [20]. Some studies have also reported that the amylose affects the pasting properties in millet [20]. However, few studies have investigated the starch physicochemical properties in sorghum varieties.

Accordingly, in this study, we selected six Chinese sorghum varieties and compared their chemical compositions, physicochemical, thermal, and pasting properties of starches. This study also aimed to provide useful information for not only the application of sorghum starches in the food industry but for efficient and productive intercropping model to develop sorghum based intercropping. Moreover, our findings will led a foundation for selection of appropriate sorghum cultivars for intercropping models to achieve the sustainable goal and farm based efficient cropping models for local small scale farmers.

## 2. Results and Discussion

### 2.1. Chemical Analysis

The chemical compositions of sorghum starches are shown in Table 1. The fat content did not significantly differ (*p* > 0.05). The contents of protein, and starch in sorghum starches were 0.80–1.20% and 85.26–96.42%, respectively. Jinnuo 3 had the highest protein content compared with the percentages in other sorghum starches. The high starch content in these six starches showed that the Chinese sorghum varieties were ideal starch resources. The findings on fat and protein contents in sorghum starches were low, which indicated that the extracted starches were pure. The contents of amylose in Jinza 34, Liaoza 19, Jinnuo 3, Jiza 127, Jiniang 2, and Jiaxian were 17.61%, 22.12%, 8.60%, 26.90%, 8.30%, and 20.47%, respectively. These results indicate that the starches of Jinnuo 3 and Jiniang 2 had the lowest amylose content than the others.

### 2.2. Microscopy Analysis of Sorghum Starches

The morphology of different sorghum starches were observed via polarized light examination and scanning electron microscopy (Figure 1). Polarized light examination was useful in observing the granules of different sorghum starches; these granules were black in the middle and showed the characteristic “Maltese cross”. Ambigaipalan et al. [21] reported that sorghum starch granules are like crystal balls and “Maltese cross”. All these starches showed the different sorghum starch granules, which had various shapes, such as round, polygonal, and regular. Besides various shapes, there were honeycomb structures found on the surfaces of all starch granules of sorghum, which may probably caused by the erosion by alkali steeping in the process of starch isolation [22,23]. The starch granules of Jinza 34, Jinnuo 3, Jiza 127, Jiniang 2, and Jiaxian were intermittently polygonal in the corners.

Fiji ImageJ program is useful as a practical and research tool in observing the physics structure of materials and scientific images [24,25]. The granule size distribution histogram and line fit of Lognormal single peak function were shown in Figure 2. According to the Fiji ImageJ software results, the average size of granules of Jinza 34, Liaoza 19, Jinnuo 3, Jiza 127, Jiniang 2, and Jiaxian were 4.31, 4.18, 4.44, 4.34, 4.30, and 4.22 μm, respectively. The average granule sizes of Jinnuo 3 (4.44 μm) were larger than those of other granules. The size of sorghum starch granules ranged from 4 to 20 μm [26]. Our results, at least, suggest a general idea regarding the sample’s granules size. It can generally be noted that samples with smaller granule sizes have smaller crystallites sizes. This can be attributed to the fact that, each sample granule was constructed by many crystallites [24,27].

### 2.3. Flow Cytometry Analysis

The flow cytometry is currently widely used as an effective method for the classification of starch granules. In this study, starches were isolated from six sorghum cultivars and initially classified into various subgroups by flow cytometry and then collected through flow sorting to observe their morphology by SEM [23] and particle-size distribution by laser particle distribution analyzer [17,28] or Fiji ImageJ program [24,25]. Zhang et al. [29] used the abovementioned methods in maize starch analysis. We anasomlyzed plots of forward-scattered light (FSC) and side-scattered light (SSC) and 1-aminopy-3, and -6 and 8-trisulfonic acid (APTS) and SSC, and we obtained a histogram of unstained and APTS-stained starches in order to determine characteristics of the starch granules (Figure 3). The SSC, FSC, and APTS showed the overall integral structures’ complexity and fluorescence intensity. The six starch granules of sorghum were divided into two particle percentage of subgroups, P1 and P2. However, the particle percentage values of the same group were different among the six starch granules. Liaoza 19 (P1 = 66.5%) contained larger and more complex granules than the rest of other sorghum starches. The smallest starch granules were found in Jinnuo 3. The P2 subgroup with the smallest and simplest granules contained fewer stained granules, which may have been due to the presence of few impurities in the starch. The results revealed that flow cytometry provides good results and efficiently classifies starch granules.

### 2.4. X-ray Diffraction Analysis

The X-ray diffraction diagrams of the starches are shown in Figure 4A. The starches of Jiaxian, Jiniang 2, Jiza 127, Jinnuo 3, Liaoza 19, and Jinza 34 showed the pattern type “A” and had diffraction peaks at 15° and 23° 2θ and constant double peaks at 17° and 18° 2θ, which is consistent with the results in other studies [26,28]. In this study, all sorghum starches exhibited similar X-ray diffraction patterns. The relative crystallinity was 36.32%, 35.00%, 33.59%, 33.79%, 30.76%, and 33.75% for Jinza 34, Liaoza 19, Jinnuo 3, Jiza 127, Jiniang 2, and Jiaxian, respectively (Figure 4B). In our studies, Jinza 34 had the highest relative crystallinity (36.32%), whereas Jinnuo 3 starch had the lowest RC (33.59%). RC was affected by many factors (including crop biological origin, amylose and amylopectin contents, crop growth and development conditions, and maturity of parent plant at harvest time) [17,20,23].

### 2.5. Light Transmittance and Water Holding Capacity of Sorghum Starch

The light transmittance of starch, a one of the starch paste parameters, and the manufactured of starch-based products with high quality light transmittance are well known among popular amongst the consumers. Figure 5A shows that Jinnuo 3 and Jiniang 2 had higher light transmittance, but were extremely low in amylose content, the results which was consistent with some studies indicating that an increase in amylose content will decrease transparency of starch paste [20]. WHC, an important parameter in determining of swelling, water against the gravity of starches, and the degree of damage of starch granules [30,31]. Figure 5B shows significant difference in the WHC of sorghum starches. Jiniang 2 and Jinnuo 3 had highest WHC values of 105.96% and 99.04%, while Jiaxian had lowest WHC value at 81.19%.

### 2.6. Water Solubility and Swelling Power of Sorghum Starch

The swelling power and water solubility parameters are shown in Table 2. The ratio varies according to the water solubility index and swelling power at temperatures of 30, 50, 70, and 90 °C and increases with temperature. The solubilized components of the granules were amylose during swelling. Amylose starch is more soluble compared with lower and higher starches than amylose granules. The solubility of Jiaxian at 30, 70, and 90 °C was much higher (*p* < 0.05) than that of the other sorghum varieties. The solubility of Liaoza 19 was relatively higher (*p* < 0.05) than that of the other sorghum varieties at 50 °C. Table 2 shows that the swelling power of Jinnuo 3 and Jiniang 2 sorghum varieties were higher (*p* < 0.05) than that of the other varieties at 30 and 50 °C. The swelling power of Jiza 127 sorghum variety was higher (*p* < 0.05) than that of the others at 70 and 90 °C. A rapid rise in solubility (%) and swelling power (g/g) was found from 70 to 90 °C, at which gelatinization occurred. Lin et al. [15] reported that starch swelling and solubility occur with the disruption of the structure caused by the breaking of hydrogen bonds between the water molecules and the exposed hydroxyl group of amylose and amylopectin. In this study, the swelling power and solubility were different between sorghum, which may be due to the amylose content. Uarrota et al. [32] reported that the swelling power and water solubility were affected by amylose, amylopectin, and granule size. Amylose inhibits the swelling power and water solubility of starches.

### 2.7. Pasting Properties of Starch by Application of Peak Viscosity

The pasting properties of different sorghum starches are shown in Table 3. Pasting temperature (PTM) and peak time (PT) ranged from 75.10 °C to 81.65 °C and from 3.56 min to 4.33 min, respectively. Higher PTM and PT values show that starch is more difficult to gelatinize. The peak viscosity (PV) is the maximum viscosity at which starch granules start to gelatize until it cools. The PV ranged from 3467.50 to 4994 mPa∙s, with the Jinza 34 starch showing the highest PV, and Jiniang 2 and Jinnuo 3 demonstrated the lowest. Trough viscosity (TV) ranged from 973.50 to 1591.00 mPa∙s, with Jiniang 2 starch having the highest TV, and Jinza 34 had the lowest. Breakdown (BD) is an indicator of the degree of granule disintegration and reflects the heat resistance of the starch; BD showed stronger heat resistance [28]. Breakdown (BD) ranged from 1964.00 to 4013.50 mPa∙s, with Jinza 34 starch having the highest BD, and Jinnuo 3 and Jiniang 2 starches show the lowest. The highest PTM and PT and the lowest PV and BD in Jinnuo 3 and Jiniang 2 starch showed strong cohesion and high thermal stability in the starch granules. In contrast, PV and BD of Jinza 34 starch were the highest of the six starches. Final viscosity (FV) and setback (SB) ranged from 2191.50 mPa∙s to 3476.50 mPa∙s and from 600.50 mPa∙s to 2037.00 mPa∙s, respectively. High FV an SB indicate low constancy and tendency to retrogradate [28]. Jiza 127 and Jiaxian sorghum starches showed the highest FV and SB, and Jinnuo 3 and Jiniang 2 had the lowest. This result indicated that Jinnuo 3 and Jiniang 2 had good stability. Jinza 34 starch showed special pasting properties, which was consistent with its starch paste properties.

### 2.8. Differential Scanning Calorimetry

The gelatinization transition temperature start (To) ranged from 68.04 to 76.11 °C; peak (Tp) ranged from 71.34 to 80.77 °C; completion (Tc) ranged from 79.14 to 88.51 °C; enthalpy for gelatinization (ΔH) ranged from 5.94 to 13.38 J/g; and gelatinization temperature ranged from 10.9 to 12.4 °C (Table 4). The high gelatinization temperature showed an indication of a higher crystalline structure in the cowpea starches that were evidenced by their higher crystallinity [33]. The process of gelatinization enthalpy energy is required to dissolve starch granules [28]. Jinnuo 3 showed the highest To, Tp, Tc, and ΔH. The ΔH of Jiza 127 was lower than the ΔH values of the other granules with crystal line shapes.

### 2.9. Principal Component Analysis (PCA) and Hierarchical Cluster Analysis (HCA) of Starches

PCA and HCA were carried out in this study to better understand the characteristics of sorghum starches (Figure 6). PC1 and PC2 explained 69.68% and 19.11% of the total variance, respectively. The PCA score plot (Figure 6A) shows the similar properties of Jiniang 2 and Jinnuo 3, which are greatly affected by comparable and relatively high values of swelling power (at 30 and 50 °C), pasting properties (TV, PT, and PTM), and thermal properties (To, Tp, Tc, and ∆H). The PCA loadings (Figure 6B) indicate different correlations in terms of the physicochemical properties of the analyzed starches. Jiniang 2 and Jinnuo 3 showed a highly negative correlation with other starches. Furthermore, the amylose content was highly negatively correlated with TV, PT, PTM, SP-30, SP-50, To, Tp, Tc, and ∆H and positively correlated with SP-90 and SP-70 (*p* < 0.05).

The HCA was performed on the basis of physicochemical properties of starches to compare the relationship of different sorghum varieties (Figure 6C). Cluster analysis depicted 47–87% dissimilarity among six sorghum varieties and was classified into two clusters. Cluster (I) was the largest cluster, which is composed of four sorghum varieties, namely, Liaoza 19, Jinza 34, Jiaxian, and Jiza 127. These four sorghum starches had higher amylose content, water solubility, peak viscosity, BD viscosity, final viscosity, and setback viscosity. Cluster (II) consisted of two sorghum varieties, including Jiniang 2 and Jinnuo 3. These two varieties were the most divergent sorghum starches. Cluster (II) had higher swelling power values at 30 and 50 °C, and TV, PT, and PTM, and higher thermal properties, which were consistent with the result of cluster analysis. Selecting varieties from both clusters may be useful in developing industrial processes for sorghum.

### 2.10. Heat Map Correlation Analysis of Starches

A heat map was created to show further correlation coefficients among amylose contents, starch contents, relative crystallinity, pasting, and thermal properties of sorghum starches. In the present study, amylose content had a significant positive correlation with SB, FV, BD, and PV, whereas a negative correlation was noted among Tp, Tc, and To in the starch samples (Figure 7). Similar correlation coefficients of relative crystallinity and starch content were found between pasting and thermal properties of starch samples.

## 3. Materials and Methods

### 3.1. Materials

Six varieties of sorghum, namely, Jinza 34, Liaoza 19, Jinnuo 3, Jiza 127, Jiniang 2, and Jiaxian, were grown in the Experimental Fields of the Northwest Agriculture and Forestry University, Yulin, Shaanxi (37°56’26″ N, 109°21’46″ E), China in 2021. The same sowing and environmental growth conditions were used to all planting sorghum materials. The seeds were separated from the impurities and other inert-materials for further use.

### 3.2. Starch Extraction

Starch was extracted using the alkaline steeping method of Singh et al. [34] and Gao et al. [35] with some modifications. Approximately 100 g of sorghum grain was immersed in 200 mL of NaOH (0.25% *w*/*v*) at 4 °C for 24 h. The soaked grains were washed, the shells were removed, and the grains were crushed into flour for 3 min with a grinder (FW-100D, Tianjin Xinbode Co., Ltd., Tianjin, China). The suspension was filtered through 100 and 200 mesh screens, and the residues on the screens were washed with water. Grinding and filtering were performed three times. After cleaning, the residues were discarded, and filtration was performed for 1 h. The filtrate was rotated at 4000 rpm. The gray protein-rich top layer was removed with a spatula for 10 min. The contaminants were easily separated by adding 0.3% NaOH and centrifuged at 4000 rpm for 10 min. The procedure was performed with a wash tank containing double-distilled water, and centrifugation was again performed at 4000× *g* for 10 min. Washing and centrifugation were repeated several times until the top starch layer was clear. The final starch was dried in a fume hood for 24 h at 40 °C, and a 100-mesh sieve was used to clean the lumps. The extracted starch was stored in an air-tight container for further characterization.

### 3.3. Chemical Analysis of Sorghum Starches

Fat was obtained through Soxhlet extraction, and the solvent was petroleum ether with a boiling point of 60–90 °C. The protein was obtained using Kjeldahl method, and the content of protein (%) was calculated from the nitrogen content (N × 6.25). The total starch contents were obtained via the anthrone spectrophotometer method and calculated by multiplying the glucose concentration with the conversion factors [26]. The amylose contents of the sorghum were measured according to a method of Yang et al. [26]. The absorbance of the solutions of different sorghum varieties was measured with the Blue Star B spectrophotometer (Lab Tech Ltd., Beijing, China). The amylose content was calculated depending on the measured absorbance according to the standard curve developed using amylose Sigma-Aldrich blends (St. Louis, MO, USA).

### 3.4. Polarized Light Microscopy Analysis

A suspension of starch 10% (*w*/*v*) and 50% glycerol was observed under polarized light. An Olympus BX53 polarized light microscope (Olympus, Tokyo, Japan) was used. The suspension was photographed with a CCD camera (Olympus DP72, Tokyo, Japan).

### 3.5. Microscopy Analysis

The sample was sprayed with gold and palladium in a ratio of 60:40 and observed with a scanning electron microscope (S4800, Hitachi, Tokyo, Japan). The granule size distribution and average diameter of starch granules was observed by using of Fiji ImageJ software (ImageJ, US. NIH, Bethesda, MD, USA).

### 3.6. Flow Cytometric Analysis

The complexity and sizes of the starch granules were determined via flow cytometry (BD FACSAria TM III, Franklin Lakes, NJ, USA), according to the method of Zhang et al. [29].

### 3.7. X-ray Diffraction Analysis

The structure of the crystal test of starch was measured with an X-ray diffractometer (D/Max2550VB+/PC, Rigaku Corporation, Tokyo, Japan). The Cu Ka X-ray wavelength was operated at an accelerating voltage of 40 kV and a current of 100 mA. The samples were scanned over the range of 5°–50° 2θ with a step size of 0.02° [36]. The relative crystallinity (RC) of the starch granules was calculated as described by Rabek [37] using the following equation: RC (%) = (Ac/(Ac + Aa)) × 100, where Ac is the crystalline area; and Aa is the amorphous area on the X-ray diffractograms.

### 3.8. Light Transmittances and Water Holding Capacity

According to the method described by Yang et al. [23], the light transmittances of starch was measured with spectrophotometer (Blue Star B, Lab Tech Ltd., Beijing, China). Approximately 0.5 g of starch and 10 mL of H_2_O were mixed in a centrifuge tube and then placed in a 100 °C boiling water for 30 min. Then the mixture was centrifuged again under 5000 r/min for 15 min and the supernatant was discarded. The water holding capacity (WHC) of the starch granules was calculated as described by Yang et al. [38] using the following equation: WHC (%) = ((Weight of gelatinized starch − 0.5)/0.5) × 100.

### 3.9. Water Solubility and Swelling Power of Sorghum Starch

The swelling power and solubility in water were measured using the method of Uarrota et al. [32]. A 500 mg starch suspension and 20 mL of distilled water were heated in a water bath at 30, 50, 70, and 90 °C and mixed for 30 min. The suspension was cooled to 25 °C and rotated at 3000 rpm for 20 min. The supernatant was carefully decanted, and 10 mL of the residue was collected, transferred to a Petri dish (known weight), dried (60 °C, hold for 12 h), cooled, and weighed to determine the solubility index. The swelling power (%) was calculated in the ratio of the wet mass of the sediment gel to the dry matter in the gel.

### 3.10. Differential Scanning Calorimetry

Starch thermal property was measured by DSC (DSC; Q 2000, TA Instruments, Wood Dale, IL, USA). Approximately 3.00 mg of the sample was placed in an aluminum crucible with 2:1 deionized water and sealed with a matching base and allowed to stand for 2 h at room temperature. The sample aluminum crucible was heated in 20 °C to 110 °C at a rate of 10 °C/min. The sealed blank aluminum crucible was used as the control. Parameters recorded were gelatinization onset, peak, conclusion, temperatures and enthalpy, and gelatinization range.

### 3.11. Pasting Properties

The pasting properties of starches measured with a rapid viscosity analyzer (RVA 4500, Perten, Hägersten, Sweden). The 3 g of dried starch was mixed with 25 mL of water and heated at 50 °C for 1 min and heated at a rate of 12 °C/min to 95 °C. The sample was kept at 90 °C for 2 min, cooled at a rate of 12 °C/min, and reserved at 50 °C for 1 min [39]. Peak viscosity, trough viscosity, breakdown viscosity, final viscosity, setback viscosity, peak time, and pasting temperature were recorded. All the viscosity parameters are expressed in millipascal-seconds (mPa∙s).

### 3.12. Statistical Analysis

For sample characterization, measurements were performed at least twice unless otherwise specified. All data are represented as the mean values ± standard deviations. Data were subjected to one-way variance and Tukey’s multiple-comparison analysis by using IBM SPSS version 20.0 (SPSS Inc. Chicago, IL, USA). A *p*-value of 0.05 was considered statistically significant. The PCA was performed on the basis of the correlation matrix. The HCA was used in determining the link among the groups, and the Ward’s cluster method and Euclidean distance as the interval measures were used. Statistical software XLSTAT new version 2020.5 was used in determining the physicochemical properties of the sorghum starches.

## 4. Conclusions

We studied the chemical compositions and physicochemical properties of starches from six sorghum varieties. All sorghum starch granules showed regular polygonal round shapes and exhibited typical “Maltese crosses”; however, they had different complex granules. The starches of Jiniang 2 and Jinnuo 3 had the lowest amylose contents than those of the other starches. These six sorghum starches all exhibited the same A-type diffraction pattern; however, the relative crystallinity of Jinza 34 was higher than the other starches. The six starch pastes had differences in water solubility, swelling power, light transmittance, and pasting and thermal properties. Jinnuo 3 and Jiniang 2 sorghum cultivars required more energy for starch gelatinization. Sorghum starch had good stability and is suitable for use as a frozen food thickener or food additive, and as a raw material for porridge, couscous, and mayonnaise. Under marginal lands with low input conditions, successful cultivation of crops is a common challenge and sorghum-based intercropping is one of the promising options for agricultural sustainability and rural livelihood security of small scale farmers in arid, semi-arid, and water-logged regions.

## Figures and Tables

**Figure 1 plants-11-01574-f001:**
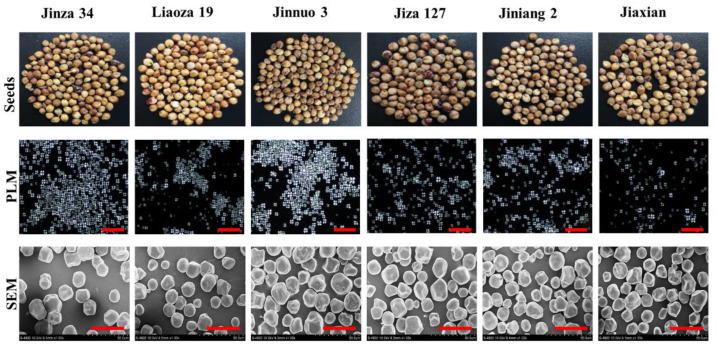
Photographs of seeds and morphologies of isolated sorghum starch granules (PLM—polarized light microscope, SEM—scanning electron microscope). Red scale bar of PLM and SEM were 200 μm and 50 μm, respectively.

**Figure 2 plants-11-01574-f002:**
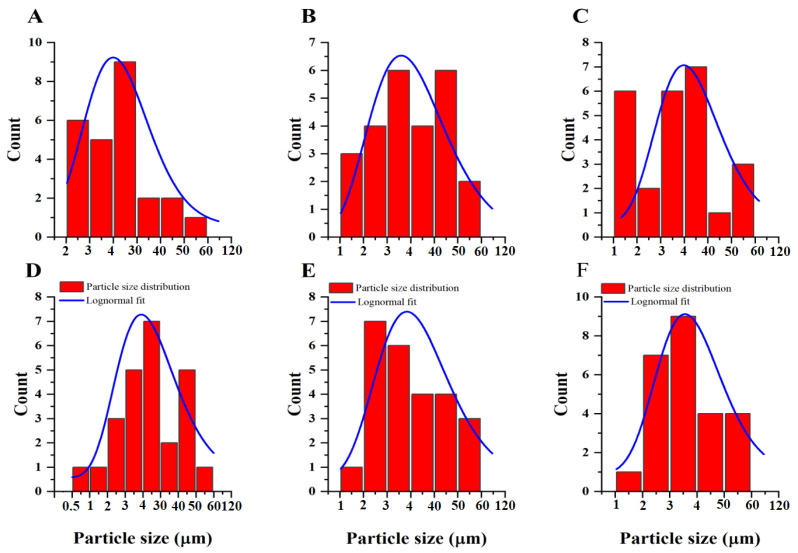
Particle size distribution analysis and Lognormal fitting of the sorghum starch granules: Jiza 34 starch granule (**A**), Liaoza 19 starch granule (**B**), Jinnuo 3 starch granule (**C**), Jiza 127 starch granule (**D**), Jiniang 2 starch granule (**E**), and Jiaxian starch granule (**F**).

**Figure 3 plants-11-01574-f003:**
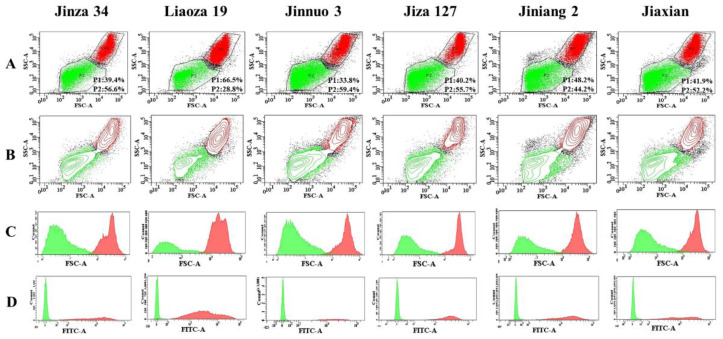
Bivariate flow cytometric histograms of six sorghum starches: Side scattered-forward scattered (SSC-FSC) image (**A**) (Paad1, P2—particle percentage of subgroups in different starches obtained by flow cytometry analysis), fluorescence image (**B**) (SSC-A—side scattered light; FSC-A—forward scattered light), histogram of unstained starch (negative control) (**C**) (FSC-A—forward scattered light), histogram of 1-aminopyrene-3,6,8-trisulfonic acid (APTS)-stained starch (**D**) (FITC-A—fluorescein isothiocyanate).

**Figure 4 plants-11-01574-f004:**
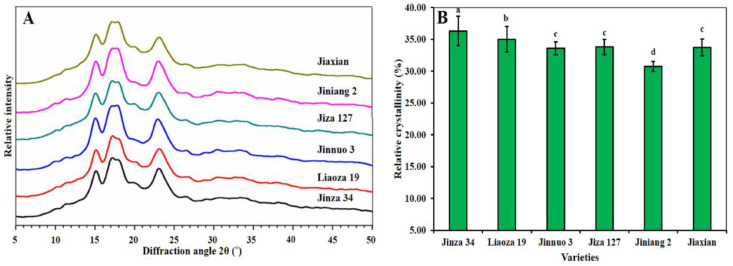
X-ray diffraction (XRD) patterns (**A**) and relative crystallinity (RC%) (**B**) of the starches. The data showed the mean of three replications, and error bars are standard deviations. Different letters (a–d) indicate there were significant differences (*p* < 0.05) in the LSD means comparisons between the treatments mean.

**Figure 5 plants-11-01574-f005:**
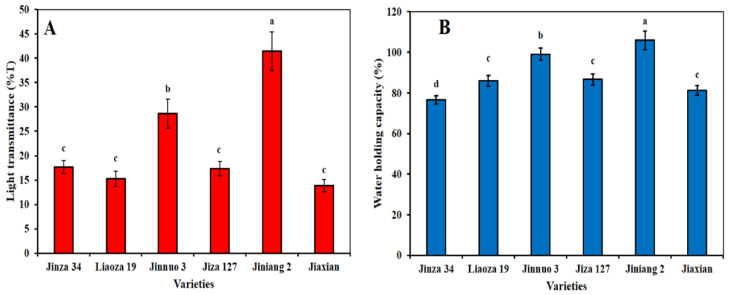
Light transmittance (%T) (**A**) and water holding capacity (WHC%) (**B**) of the starches. The data showed the mean of three replications, and error bars are standard deviations. Different letters (a–d) indicate there were significant differences (*p* < 0.05) in the LSD means comparisons between the treatments mean.

**Figure 6 plants-11-01574-f006:**
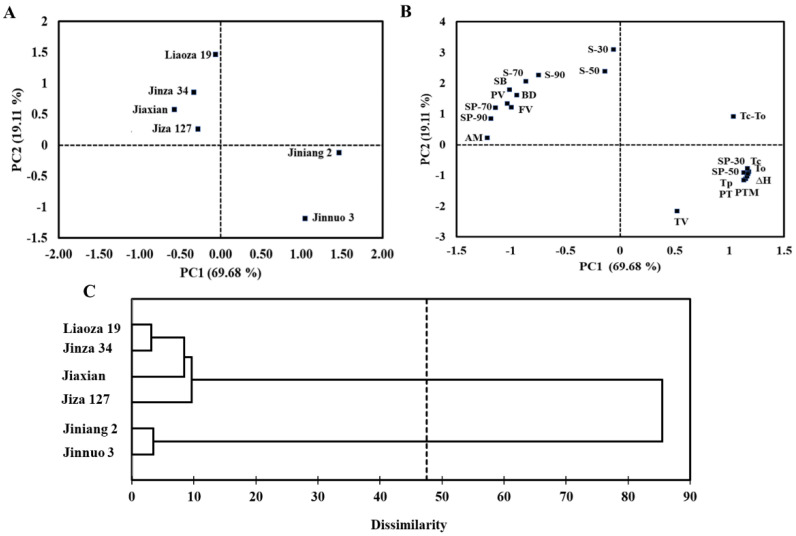
Principal components analysis score (**A**) and loading plot (**B**) of physicochemical properties for six sorghum starches showing the first two principal components (PC—principal component; S—solubility; SP—swelling power; AM—amylose content; PV—peak viscosity; TV—trough viscosity; BD—breakdown viscosity; FV—final viscosity; SB—setback viscosity; PTM—pasting temperature; PT—peak time; To—gelatinization onset temperature; Tp—gelatinization peak temperature; Tc—gelatinization conclusion temperature; ΔH—gelatinization enthalpy; Tc-To—gelatinization range). The dendrogram of hierarchical cluster analysis was based on the physicochemical parameters of six sorghum starches (**C**).

**Figure 7 plants-11-01574-f007:**
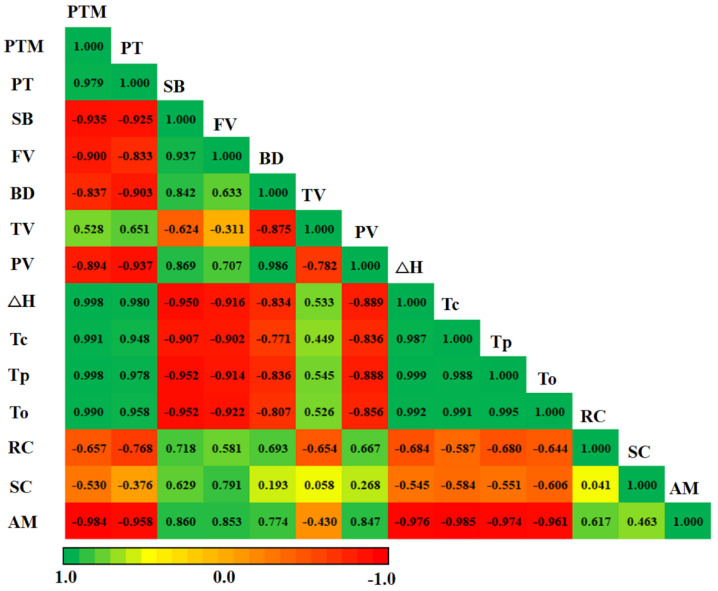
Pearson’s correlation coefficients between structural and physicochemical properties of the starch samples (PV—peak viscosity; TV—trough viscosity; BD—breakdown viscosity; FV—final viscosity; SB—setback viscosity; PTM—pasting temperature; PT—peak time; To—gelatinization onset temperature; Tp—gelatinization peak temperature; Tc—gelatinization conclusion temperature; ΔH—gelatinization enthalpy; RC—relative crystallinity; SC—starch content; AM—amylose content). The numbers in each field represent the correlation extent; the color represents significant correlation *(p* < 0.05); the deeper the color of the field, the more significant the correlation (*p* < 0.01). The green color means a positive correlation, and the red color means a negative correlation.

**Table 1 plants-11-01574-t001:** Chemical compositions of sorghum starches.

Varieties	Fat (%)	Protein (%)	Starch (%)	Amylose (%)
Jinza 34	0.01 ± 0.01	0.86 ± 0.19 ^b^	88.82 ± 0.60 ^b^	17.61 ± 2.10 ^c^
Liaoza 19	0.01 ± 0.00	0.89 ± 0.05 ^b^	90.24 ± 1.23 ^b^	22.12 ± 2.02 ^b^
Jinnuo 3	0.02 ± 0.01	1.20 ± 0.09 ^a^	85.26 ± 0.30 ^c^	8.60 ± 0.62 ^d^
Jiza 127	0.01 ± 0.00	0.87 ± 0.30 ^b^	89.98 ± 0.90 ^b^	26.90 ± 1.95 ^a^
Jiniang 2	0.01 ± 0.01	0.86 ± 0.20 ^b^	89.94 ± 0.94 ^b^	8.30 ± 0.61 ^d^
Jiaxian	0.01 ± 0.00	0.80 ± 0.20 ^c^	96.42 ± 1.30 ^a^	20.47 ± 3.39 ^b^

Data are represented as the mean ± standard deviations. For each column, values not displaying the same letter are significantly different (*p* < 0.05).

**Table 2 plants-11-01574-t002:** Swelling power and solubility of different sorghum starches at different temperatures.

Varieties	Solubility (%)				Swelling Power (g/g)			
	30 (°C)	50 (°C)	70 (°C)	90 (°C)	30 (°C)	50 (°C)	70 (°C)	90 (°C)
Jinza 34	0.62 ± 0.04 ^b^	0.87 ± 0.24 ^b^	3.80 ± 0.24 ^b^	4.78 ± 0.19 ^b^	1.89 ± 0.08 ^c^	1.93 ± 0.05 ^b^	10.83 ± 0.30 ^c^	11.01 ± 0.83 ^c^
Liaoza 19	0.44 ± 0.08 ^c^	0.93 ± 0.12 ^a^	3.00 ± 0.10 ^c^	3.53 ± 0.23 ^c^	1.89 ± 0.03 ^c^	1.92 ± 0.05 ^b^	11.30 ± 0.40 ^b^	11.47 ± 0.44 ^b^
Jinnuo 3	0.15 ± 0.10 ^e^	0.22 ± 0.15 ^f^	0.73 ± 0.13 ^d^	0.91 ± 0.19 ^e^	2.19 ± 0.06 ^a^	2.21 ± 0.15 ^a^	3.11 ± 0.42 ^e^	6.54 ± 0.15 ^d^
Jiza 127	0.22 ± 0.04 ^d^	0.37 ± 0.19 ^e^	3.20 ± 0.04 ^c^	4.02 ± 0.31 ^c^	1.83 ± 0.13 ^d^	1.89 ± 0.07 ^c^	12.41 ± 0.29 ^a^	12.72 ± 0.25 ^a^
Jiniang 2	0.46 ± 0.10 ^c^	0.62 ± 0.04 ^c^	0.84 ± 0.12 ^d^	2.33 ± 0.19 ^d^	2.18 ± 0.09 ^a^	2.26 ± 0.11 ^a^	3.74 ± 0.73 ^d^	6.04 ± 0.22 ^d^
Jiaxian	0.78 ± 0.03 ^a^	0.44 ± 0.14 ^d^	5.07 ± 0.00 ^a^	6.22 ± 0.23 ^a^	1.93 ± 0.04 ^b^	2.04 ± 0.15 ^ab^	10.67 ± 0.20 ^c^	11.02 ± 0.68 ^c^

Data are represented as the means ± standard deviations. For each column, values not displaying the same letter are significantly different (*p* < 0.05).

**Table 3 plants-11-01574-t003:** Pasting properties of sorghum starches.

Varieties	Pasting Properties						
	PV (mPa∙s)	TV (mPa∙s)	BD (mPa∙s)	FV (mPa∙s)	SB (mPa∙s)	PT (min)	PTM (°C)
Jinza 34	4994.00 ± 226 ^a^	973.50 ± 44 ^d^	4013.50 ± 171 ^a^	2807.50 ± 70 ^b^	1834.00 ± 25 ^a^	3.70 ± 0.0 ^b^	77.22 ± 0.6 ^b^
Liaoza 19	4468.5 0 ± 88 ^bc^	1185.50 ± 00 ^c^	3283.00 ± 87 ^b^	2880.00 ± 106 ^b^	1694.50 ± 50 ^a^	3.67 ± 0.0 ^c^	76.35 ± 0.6 ^b^
Jinnuo 3	3467.50 ± 77 ^d^	1503.50 ± 27 ^ab^	1964.00 ± 49 ^d^	2192.50 ± 51 ^c^	689.00 ± 24 ^b^	4.20 ± 0.1 ^a^	81.65 ± 0.0 ^a^
Jiza 127	4861.00 ± 302 ^b^	1396.00 ± 114 ^b^	3465.00 ± 188 ^b^	3117.00 ± 292 ^a^	1721.00 ± 78 ^a^	3.56 ± 0.0 ^c^	75.10 ± 0.0 ^c^
Jiniang 2	3484.50 ± 20 ^d^	1591.00 ± 50 ^a^	1893.50 ± 14 ^d^	2191.50 ± 20 ^c^	600.50 ± 30 ^b^	4.33 ± 0.0 ^a^	81.60 ± 1.2 ^a^
Jiaxian	4292.50 ± 211 ^c^	1439.50 ± 53 ^ab^	2853.00 ± 158 ^c^	3476.50 ± 379 ^a^	2037.00 ± 96 ^a^	3.76 ± 0.0 ^b^	76.40 ± 0.6 ^b^

PV—peak viscosity; TV—trough viscosity; BD—breakdown viscosity; FV—final viscosity; SB—setback viscosity; PTM—pasting temperature; PT—peak time; mPa∙s—millipascal-seconds. Data are represented as the means ± standard deviations. For each column, values not displaying the same letter were significantly different (*p* < 0.05).

**Table 4 plants-11-01574-t004:** Thermal properties of sorghum starches.

Varieties	Thermal Properties				
	To (°C)	Tp (°C)	Tc (°C)	ΔH (J/g)	Tc-To (°C)
Jinza 34	70.22 ± 3.1 ^c^	74.01 ± 3.0 ^b^	82.12 ± 2.2 ^d^	8.21 ± 1.9 ^b^	11.9 ± 0.2 ^b^
Liaoza 19	68.26 ± 3.1 ^d^	72.56 ± 3.0 ^c^	79.14 ± 2.4 ^d^	7.28 ± 2.2 ^bc^	10.9 ± 0.1 ^c^
Jinnuo 3	76.11 ± 0.1 ^a^	80.77 ± 0.0 ^a^	88.51 ± 0.6 ^a^	13.38 ± 0.5 ^b^	12.4 ± 0.3 ^a^
Jiza 127	68.04 ± 0.2 ^d^	71.34 ± 0.3 ^d^	77.70 ± 0.9 ^e^	5.94 ± 0.5 ^c^	9.7 ± 0.0 ^d^
Jiniang 2	75.74 ± 4.5 ^b^	81.00 ± 4.3 ^a^	87.47 ± 5.2 ^b^	13.68 ± 2.6 ^a^	11.7 ± 0.2 ^b^
Jiaxian	68.28 ± 4.5 ^d^	72.47 ± 4.4 ^d^	79.52 ± 6.1 ^c^	7.02 ± 3.7 ^bc^	11.2 ± 0.2 ^b^

To—gelatinization onset temperature; Tp—gelatinization peak temperature; Tc—gelatinization conclusion temperature; ΔH—gelatinization enthalpy; Tc–To—gelatinization range. Data represent are means ± standard deviations. For each column, values not displaying the same letter are significantly different (*p* < 0.05).

## Data Availability

All obtained data are enclosed with this manuscript.

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
