# Peer review of "Integrated Starches and Physicochemical Characterization of Sorghum Cultivars for an Efficient and Sustainable Intercropping Model"

_plants, 2022, doi:10.3390/plants11121574_

Round 1

Reviewer 1 Report

correct led to lead in introduction

also, what is intercropping? perhaps can briefly define....

Reviewer 2 Report

I have reviewed the manuscript titled: Integrated starches and physicochemical characterization of sorghum cultivars for efficient sustainable intercropping model

This article aims to evaluate the physicochemical properties of starches from 6 sorghum varieties and their chemical compositions. The information of this work is useful and relevant and there are many sorghum starch physicochemical information of the manuscript that could be adapted by agriculture processing and non-food industry especially for cereal scientist and processor in the future. I think the manuscript is acceptable after minor revision. Although, the article is not innovative, it contains original and interesting basic information for sorghum processing and sustainable intercropping sorghum cultivars. Abstract is well written upon and the shape, water solubility, the flow cytometry, RVA parameters,starch type diffraction pattern, amylose content and statistic analysis of 6 sorghum starch varieties are mentioned and evaluated. Introduction is well addressed including sorghum grown regions, properties of high phenolic compounds of Chinese sorghum as food stuff. The major components, amylose and amylopectin of starch are introduced and how their ratio in sorghum, maize, millet influences the functional properties for cereal food. The information could be used for selection of appropriate sorghum varieties for intercropping sorghum for food and feed industries in this study.

Materials and methods were well described at starch extraction,chemical, differential scanning calorimetry, polarized light microscopy, microscopy,flow cytometric, X-ray diffraction analysis and operated conditions. Light transmittances, water holding capacity, water solubility, swelling power, RVA pasting properties of 6 sorghum starch cultivars were conducted. All results were subjected to ANOVA, Tukey’s multiple-comparison analysis, principal components analysis and hierarchical cluster analysis.

This article would be improved if the authors revised two cited references at pages 4 & 11 and removed the letter symbol for fat (%) of Table 1 at page 3. The reference number 10 should give abbreviated journal name (Food Hydrocoll.) as attached revised suggestion file. The particle size of Fig. 2 seems in range of 20-120 micrometer not consisting with 4 micrometer in the text at page 4.

I am not a native English speaker. The manuscript seems have no major mistakes are detected and the manuscript can be easily understood. The results are well discussed.

I enjoyed reading this manuscript; the needs of special groups of cereal growing, processing and property evaluation. This manuscript presents some basic and interesting data.

Date of this review

24 May 2022 23:31

Reviewer 3 Report

Comments to Author:

The manuscript by Htet et al. analyzed the starch and physiochemical properties of sorghum. This paper is interesting. However, despite an adequate data, I have some concerns about the abstract and conclusion that authors have made. On the basis of my careful study of this manuscript, I recommend it minor revision.

Abstract.

Abstract is very important part of each manuscript and it is transferring in the scientific base of data instead of full paper. For this reason, abstract deserves to have more consideration. The abstract is written too general (same as results). Also, in abstract authors didn’t give information about the most important results. Abstract should describe research context, briefly introduces the research method and states the research outcome and future expectation.

Keywords: I suggest the author not to include those words that are present in the title.

Introduction:

Introduction generally gives us a detailed description of starch and its industrial applications, which I think makes us a relatively complete understanding and knowledge of starch.

Materials and Methods

1.     How many replications were used for flow cytometry? Author has indicated in statistical analysis but is not clear. Three measurements or replications?

Results and Discussion

1.     Page 2, Line 86. not clear, please re-write this line “A significant difference of amylose…….”.

2.     Page 3, Line 108. What does it mean? Please re-write. The shapes of the sorghum starch granules slightly varied among the other sorghum starches…………

3.     Page 4, Line 131. Has anyone used flow cytometry before to measure the granules size? If yes, please discuss properly.

The discussion discusses the findings discovered in the paper, including explaining how the findings come out and interpreting the results in more details, which gives us a supplementary understanding of the results obtained in the paper. However, I found that the discussion of the results is very limited and notably based on the comparison of the results observed in the current study with the ones reported in the literature. Interpretation and integration of the results obtained, as well as a clear discussion of the relevance of the results found, are all lucking. This will help to highlight what new contribution this research brings.

Finally, authors are urged to seek assistance with standard English.
